# Bone Augmentation and Simultaneous Implant Placement with Allogenic Bone Rings and Analysis of Its Purification Success

**DOI:** 10.3390/ma12081291

**Published:** 2019-04-19

**Authors:** Bernhard Giesenhagen, Nathalie Martin, Ole Jung, Mike Barbeck

**Affiliations:** 1Private Practice, 34131 Kassel, Germany; info@giesenhagen-kassel.de; 2Botiss Biomaterials, 15806 Zossen, Germany; nathalie.martin@botiss.com; 3Section for Regenerative Orofacial Medicine, Department of Oral and Maxillofacial Surgery, University Hospital Hamburg-Eppendorf, 20251 Hamburg, Germany; ol.jung@uke.de; 4BerlinAnalytix GmbH, 12109 Berlin, Germany

**Keywords:** alveolar bone loss, alveolar bone regeneration, bone ring technique, freeze-dried bone allograft (FDBA), sinus augmentation freeze-dried bone allografts (FDBA), purification

## Abstract

The main objective of this manuscript was to demonstrate the use of freeze-dried bone allografts (FDBA) by means of a technique of simultaneous bone augmentation and implant placement (“Bone Ring Technique”) in different indications, i.e., ridge reconstruction and sinus floor elevation procedure with a maxillary bone height of less than 4 mm. Moreover, cases with an up to 3-year follow-up were chosen to analyze the techniques of mid-term clinical success. Finally, the purification success of the FDBA was analyzed by means of established scanning electron microscopic (SEM) and histological methods. The FBDA bone ring was applied in three different patients and indications and presented; the healing success was analyzed on the basis of radiographical and clinical images. For analysis of the purification of the allogeneic bone, previously established histological methods and scanning electron microscopy (SEM) were applied. All analyzed patient cases showed that the FDBA-based bone ring was fully integrated into newly built alveolar bone. Furthermore, the observations revealed that the three-dimensional bone reconstructions in maxilla and mandible were stable within the observational period of up to 3 years. Altogether, the present data show that the application of the Bone Ring Technique using the FDBA rings allows for successful regeneration of alveolar bone with a predictable clinical outcome, functionality and esthetics. Moreover, the material analyses showed that the allogeneic bone tissue was free of cells or cell remnants, while the (ultra-) structure of the bone matrix has been retained. Thus, the biological safety of the FDBA has been confirmed.

## 1. Introduction

A successful dental implant placement is critically influenced by the amount of alveolar bone present at the implantation site. Sufficient volume of alveolar bone is necessary to place the implant in a stable and restoratively driven manner and to achieve a long-term predictable esthetic outcome. Therefore, bone augmentation procedures are often required. These procedures can be complex and time consuming for the patient and the clinician, depending on the location and size of the bone defect and the treatment method used.

In this context, the development of innovative biomaterials in combination with established treatment concepts has made bone harvesting more and more obsolete in recent years [1,2,3]. Still considered the ‘gold standard’, autogenous grafts are harvested from patient’s intraoral and extraoral donor sites [4,5,6,7,8]. Known drawbacks of autogenous grafts compared to the use of biomaterials include increased surgical time, costs, graft availability, complications, donor site morbidity, pain, and unpredictable resorption [5,6,7,8]. The patient acceptance of these therapies is generally low, making bone substitute alternatives more and more popular [8]. 

Different bone substitute materials have been introduced into the dental clinic including so-called “natural” materials such as allogeneic and xenogeneic grafts and synthetic bone substitutes [2,3]. Especially, synthetic materials have shown most often not to provide the optimal properties needed for complex bone regeneration of the jaw [9]. In this context, xenogeneic bone substitute materials have been developed as an alternative to autologous bone transplants [10]. These materials have shown to be biocompatible and osteoconductive and perform comparably in different clinical indications such as auto- and allografts [10]. However, their organic components that can induce immunologic tissue reactions up to implant rejections or transfer pathogens such as prions that are related with bovine spongiform encephalopathy have to be removed prior to their application [11]. For this purpose, different purification protocols are used based on varying (combinations of) physical and chemical methods such as heat treatments [12]. Although it has been revealed that this kind of bone substitute material is secure, and its hydroxyapatite-based bone matrix seems to be comparable to that of the human bone matrix, its mechanical properties might also be low due to the different processing techniques [13]. Thus, allogeneic bone grafts are the most reliable alternative to autogenous bone with comparable clinical outcomes [14]. Moreover, in a former study it has been revealed that even processed freeze-dried bone allograft (FDBA) is equivalent to autogenous bone blocks regarding their volumetric graft remodeling rates for treating single tooth defects [15,16]. Block augmentation with FDBA represents a promising option due to low block graft failure rates, minimal resorption, and high implant survival rates [17,18,19]. Furthermore, the FDBA blocks are gradually remodeled into a patient’s own bone, therefore regenerating vital bone [20,21]. 

Guided bone regeneration (GBR) defined as the use of a barrier membrane to direct the growth of new bone, has become a predictable therapeutic method used routinely [22,23]. GBR can be performed as either a one-stage (combined approach) or two-stage procedure (staged approach). The combined approach places the implant simultaneously with the bone augmentation procedure. In the staged approach, the implant placement is carried out from 6 up to 12 months after GBR surgery [1]. However, this delay on the implant placement is linked to a reduced patient acceptance. 

Different concepts and biomaterials for the one-stage approach have already been developed but none of these concepts has shown to combine a bone substitute scaffold with a primary stability with the mean of a dental implant instead of titanium screws. To overcome this issue, the so-called “Bone Ring Technique” was developed based on a single surgery for bone augmentation and simultaneous implant placement [15,16,24,25,26]. This technique includes a cylindrical FDBA bone block that is fixed in a one-step procedure by means of a dental implant (Figure 1A) [15,16,24,25,26]. Interestingly, this technique includes the surgical adaption of the implant bed to the bone graft in contrast to a variety of other bone substitute materials that are produced to fit into a bone defect as demonstrated by the following surgical steps of the Bone Ring Technique:Determine the diameter of the defect and the according bone ring with a trephine (6 or 7 mm) from the related “bone ring surgical kit”.Determine the implant position with the pilot drill from the bone ring surgical kit.Prepare the ring bed recipient site with the appropriate trephine (6 or 7 mm diameter) from a bone ring surgical kit.Straighten/decorticate the implantation bed of the bonering with the planator from the bone ring surgical kit.Reduce the height of the bone ring with a diamond disc from the bone ring surgical kit.Insert bone ring into the prepared recipient site.Prepare the implant osteotomy through the bone ring with the according drills from the implant system.Place the implant at least 3 mm in the local through the bone ring to gain primary stability of the implant.8.1 Place a Fixation Cap in case the ring has mobilityRound off the edges of the bone ring.Cover defect with a bone substitute material.Cover the graft with a barrier membrane and close the wound.

In further studies, the healing properties of the FDBA rings has already been described but there is still limited literature addressing the Bone Ring Technique [16,26]. Thus, the objective of this case series is to introduce its use in GBR procedures in its main indications. Additionally, cases with an up to 3-year follow-up were chosen to document the techniques’ mid-term clinical success.

Furthermore, it has been shown that different allogeneic bone blocks that are available on the market might contain cells or cellular remnants and it was concluded that these components could be a source for immunogenic reactions up to the rejection of the biomaterials [27]. The authors used histological and histopathological analysis methods to examine the composition of the bone blocks. Thus, the same methods were applied to analyze the composition and security of the FDBA used for the Bone Ring technique and additionally scanning electron microscopy (SEM) was conducted.

## 2. Material Analyses

### 2.1. FDBA Bone Ring

The maxgraft® bonering (botiss biomaterials GmbH, Zossen, Germany) is produced outgoing from allogeneic cancellous bone derived from the bone of femoral heads of living human donors from German, Austrian and Swiss hospitals (Figure 1A,B) [28,29]. The purification of the bone tissue is in accordance with the respective European Directives and the Austrian Tissue Safety Act and is validated by independent institutes and by the Austrian Health Ministry [28,29]. The purification process is a highly secure quality process that is in compliance with the highest quality standards that are employed when inactivating viruses and bacteria [28,29,30]. The process includes different physical and chemical purification steps, i.e., an ultrasonic-based removal of blood, cells and tissue components; chemical and oxidative cleaning steps by means of diethyl ether and ethanol at different durations; and an oxidative purification step. Finally, lyophilization and sterilization via gamma irradiation are applied to preserve the natural tissue structure [30,31].

### 2.2. Materials and Methods

To analyze the composition of the FDBA as the basis for the bone ring, three material samples from three different batches were histologically processed for further microscopic examination as previously described [27,30,32]. In brief, the samples were initially decalcified in tris-buffered 10% EDTA (Carl Roth, Karlsruhe, Germany), dehydrated in a series of increasing alcohol concentrations followed by xylol application and embedded in paraffin. After that, histological sections of 3–5 μm thickness were made using a rotation microtome (SLEE, Wetzlar, Germany). Masson-Goldner’s trichrome was stained and furthermore a histochemical method for the detection of tartrate-resistant acid phosphatase (TRAP) was used to identify potential osteoclasts [27,30,32]. For control of the quality of this staining method, a bone section was used. The histological analysis of the composition of the bone block was conducted following previously described methods [27,30,32]. Briefly, the histological slides were examined microscopically with respect to material characteristics such as the bone matrix structure and also components like collagen or cells/cell remnants using a light microscope (AxioScope A1, Zeiss, Jena, Germany) and microphotographs were made using a connected digital camera (Axiocam 105 color, Zeiss, Jena, Germany). Moreover, scanning electron microscopy (SEM) was performed to analyze the (ultra-)structure of the FDBA using a Zeiss Supra 55 scanning electron microscope (QS Nr. 57113, Carl Zeiss SMT AG, Oberkochen, Germany) with a field electron emitter. 

### 2.3. Results of the Material Analysis

The material analysis showed that the bony architecture including the trabecular structure with the typical macro- and micropores as well as the lamellar subarchitecture of the bone matrix was preserved and no cells nor cellular remnants were observable (Figure 1). Thus, the osteocyte lacunae were found empty and no cells were detected to be associated with surfaces of the matrix in the three analyzed bone blocks (Figure 1C). Slight amounts of collagen fibers were found to be associated with the outer surfaces of the bone matrix and also no related soft tissue cells were detected (Figure 1C). The surface of the bone matrix showed a fiber-like appearance that indicates its preservation status (Figure 1E).

## 3. Patient cases

### 3.1. Case 1

A 56-year-old female patient, non-smoker without major risk factors presented with a former apiectomy in frontal maxilla. Teeth #11 and 22 had fractured roots and the areas were infected. The radiological images showed severe bone loss in the frontal maxilla (#11, #12, #21 and #22). Teeth 11 and 22 already lost two-thirds of their vertical attachment. The overdenture showed some mobility and had become symptomatic, which made removal inevitable (Figure 2A,B). The treatment plan consisted of the extraction of the endodontically treated teeth (#11, #22 and #12) and the removal of the prosthetics suprastructure. Following the surgical steps, an immediate implantation of two bone ring blocks (maxgraft® bonering Ø 7mm, botiss biomaterials, Zossen, Germany) and fixation of the blocks with dental implants (Ankylos Implant, Dentsply Sirona, York) was performed in regio #11 and #22 (Figure 2C). The defect size was initially measured with the 7-mm trephine from the maxgraft® bonering surgical kit (botiss biomaterials, Zossen, Germany) (Figure 2B). Additionally, socket preservation with gingiva graft from the tuber maxillae at #12 was performed with the goal to gain papilla through ovate pontic restoration. The site was covered with a volume-stable bovine bone substitute (cerabone®, botiss biomaterials, Zossen, Germany) and a resorbable native collagen membrane made of porcine pericardium (Jason membrane, botiss biomaterials, Zossen, Germany) (Figure 2D). The post-operative check-up showed an eventless healing and healthy soft tissue 6 months after surgery (Figure 2E). At that time, the re-entry was performed and healing abutments were placed in order to shape the gingiva. The graft appeared to be well integrated into the native bone (Figure 2F). Six weeks later, the temporary crowns were integrated, and the first esthetic results were visible only 7.5 months after surgery (Figure 2G,H). The patient was highly satisfied with the result. A radiological control at 36 months after initial surgery showed stable bone around the shoulders of the implants that indicate a mid-term esthetic outcome (Figure 2I). 

### 3.2. Case 2

A 50-year-old male patient presented with multiple chronic inflammation at teeth #24, #25 and #34, endodontically treated #16 and hopeless tooth #12. The patient’s treatment options were reviewed prior to his consent to strategic extraction and implant placement. He was advised that a regenerative procedure would be necessary due to the advanced bone loss in region #12, #24–26, #35–37. His preference was an implant fixed restoration in a single-staged approach. The treatment plan was bone augmentation including sinus floor elevation in the second quadrant and implant placement in region #12, #16, #18, #24, #26, #28, #35, #37, #46, and #48. After extraction (Figure 3A,B), three of the sites (#12, #24-25 and #34) were measured with the trephine from the maxgraft® bonering surgical kit (botiss biomaterials, Zossen, Germany) 7 mm in diameter. Accordingly, all of the sites were planned to be treated with 7-mm bone ring (maxgraft® bonering, botiss biomaterials, Zossen, Germany). Tooth 12 showed palatal and buccal bone loss (Figure 2C,D). The site was prepared according to surgical protocol and the graft was fixated with a dental implant (Ankylos Implant, Dentsply Sirona, York). The posterior maxilla in the second quadrant was also treated with immediate implant placement (#24), a 7-mm bone ring fixated with a dental implant (#26) and external sinus floor elevation including implant placement (#28) (Figure 2B) and covered with a bovine bone substitute (cerabone®, botiss biomaterials, Berlin) and a collagen membrane (Jason® membrane, botiss biomaterials, Berlin) in order to prevent resorption and soft tissue inclusion (Figure 2C). The mandible was treated with a cylindrical bone block after preparing the site with a trephine and planator to create a uniform recipient site. Dental implants at #37, #46 and #47 were placed simultaneously. Eight months after surgery, the radiological image indicated well integrated bone grafts and dental implants (Figure 2D). At that time, the re-entry was performed and healing abutments integrated in order to give shape to gingiva. The area showed hard and soft tissue maturation revealing what appeared to be an osseointegrated implant; both clinically and radiographically with complete regeneration of the buccal and palatal bone (Figure 2D,E). Five weeks after re-entry, the soft tissues seemed healthy and final prosthesis were integrated (Figure 2F). Twenty-one months after surgery, the soft tissue situation and the prosthetics looked good and the patient was satisfied with the clinical outcome (Figure 2G). The 14-months post-operative radiological image showed stable implants and bone grafts (Figure 2H). The 36 months image still showed similar results (Figure 2I). 

### 3.3. Case 3

The third case presented is a sinus floor elevation with a maxillary bone height of 0.4 -1.0 mm and the presence of sinus septa, which made that case even more challenging. The female 53-year-old patient without risk factors presented with a bridge retained posterior maxilla #24–28. Due to lack of physical load on #25–27, the site was missing sufficient bone to place implants. Additionally, the teeth #23–24 had to be extracted, as they lost attachment and were mobile (Figure 4A). The patient now desired a fixed prosthetic restoration with dental implants as soon as possible. As the bone height was only 1.5 mm at #27, the conventional treatment concepts would have meant placing the dental implant in a second stage after maturation of the bone grafting material in the sinus cavity. The Bone Ring Technique allows placing the implant simultaneously, as a fixation screw, and fixates the implant crestally within the bone graft in the sinus cavity. After being properly informed, the patient decided for a single stage approach with a cylindrical bone graft for her bone augmentation procedure at #27 and implant placement including external sinus floor elevation with a bone substitute. First, tooth #23 and #24 were extracted and a dental implant (Straumann® BLT Implant, Basel, Switzerland) was placed simultaneously at #23. The sinus septa at #25 and the bone loss in the sinus #26–27 were treated with external sinus floor elevation (Figure 4B). The lateral window was opened with a round bur and the Schneiderian membrane was gently lifted with a sinus membrane elevator. At #27, a cylindrical FDBA (maxgraft® bonering Ø 7-mm, botiss biomaterials, Zossen, Germany) was placed through the window and a bovine bone substitute (cerabone®, botiss biomaterials, Berlin) was filled into the sinus cavity (Figure 4C,D). A dental implant (Straumann® BLT Implant, Basel, Switzerland) was placed from the crest into the allogenic ring (Figure 4D). In order to secure the ring and prevent the dental implant and bone ring from moving into the sinus cavity, it was fixed with a Closure and Fixation Cap (Straumann®, Basel, Switzerland) (Figure 4E) that has a wider diameter than the implant to secure it crestally. In the end, the site #25 with the septum had sufficient bone to place a dental implant (Straumann® BLT Implant, Basel, Switzerland) with primary stability. The remaining free space in the sinus cavity was filled with bovine bone substitute (cerabone®, botiss biomaterials, Zossen, Germany) (Figure 4F). The lateral window was covered with a membrane and closed free of tension. The post-op radiological control showed well positioned implants (Figure 4G). Six months after surgery, the radiological check-up indicated stable bone in the sinus; hence the implants were uncovered, and the impressions were taken 1-week after (Figure 4H,I). The patient received the final prosthetic restoration only 6.5 months after initial surgery and was very satisfied with the final outcome (Figure 4J). 

## 4. Discussion

Allogeneic bone grafts are established as a reliable alternative to autogenous bone with comparable clinical outcomes [14,15,16,17,18,19]. Amongst the different allogeneic bone substitute materials, even processed freeze-dried bone allograft (FDBA) is often selected due to its comparability to autogenous bone blocks regarding the volumetric graft remodeling rates for treating single tooth defects and the related remodeling into patient’s own new bone [14,15,16,17,18,19].

While most of the bone substitute blocks are used for two-staged implant procedures, only a small number of solutions are available for one-stage implantation combining the implantation of the tooth implant with the bone substitute [15,16,17]. The so-called “Bone Ring Technique” is one of the combined approaches as it includes one single surgery using a cylindrical FDBA bone block for bone augmentation that is simultaneously fixated by implant placement. Thereby, this technique includes the surgical adaption of the implant bed to the bone graft in contrast to a variety of other bone substitute materials that are produced to fit into a bone defect [15,16,17]. The successful application of this technique has already been described but there is still limited literature [16,17]. Thus, the objective of this manuscript was to analyze its usability in GBR procedures in its main indications. To examine the techniques’ mid-term clinical success, cases with an up to 3-year follow-up were moreover included. 

In all of the three patient’s cases, a bone augmentation procedure was necessary, and the patients chose to undergo a single-staged approach with an allogenic bone graft. The implant together with the bone ring could be placed in a prosthetically restorable manner with sufficient primary stability in all cases. Altogether, six bone defects were treated with a cylindrical-shaped FDBA and a dental implant. One of these defects was in the mandible and five defects were single tooth restorations in the maxilla, including one with need for bone augmentation in the sinus. Two of the cases were followed up 3 years after the surgery except for the sinus case, which was followed up 7 months after initial surgery and showed only successful prosthetic rehabilitation. Overall, the presented cases showed sufficient bone regeneration. At the time of re-entry, the FDBA rings in case 1 and 2 were visibly vascularized and well-integrated into newly formed bone tissue with almost no resorption. The esthetic appearance 3 years after initial surgery confirms that impression. In these cases, important factors for successful bone augmentation procedure were sufficient soft tissue mobilization and tension-free sutures to prevent dehiscence. Additionally, factors that may influence clinical outcomes such as patient anamnesis and instructions should be considered before performing GBR procedures [25]. 

Another application of the cylindrical-shaped FDBA is in external sinus floor elevation procedures. In case 3, the maxillary bone height was successfully reconstructed as shown by the radiological pictures. Although this case only showed a limited observation period, also the long-term success of this technique is assumable as the application of this technique has already been proven not only for the reconstruction of horizontal and vertical ridge defects but also for sinus floor elevation procedures [24,26]. However, a longer follow-up period is necessary to assure the success of this technique in sinus floor elevation procedures. 

A further aim of the present study was the analysis of the purification of the used FDBA bone block. In this context, it has been published that different allogeneic bone substitutes may contain cells or cellular remnants that are suspected to induce immunogenic reactions up to the rejection of a “natural” biomaterials such as the analyzed bone blocks [27]. Interestingly, the analyzed bone blocks were classified into four different groups based on their respective (ultra-) structure of the bone matrix as well as organic contents such as collagenous structures and cellular remnants. This classification ranges from a complete purification of the bone matrix with a loss of its lamellar structure up to materials that contain both the bone matrix with its origin lamellar structure and collagenous structures of the bone tissue as well as cells or cellular remnants. In the present study, we have additionally analyzed the allogeneic block maxgraft® using the same histological methods. Additionally, scanning electron microscopy has been conducted.

The results of this study part show that the bone block exhibited a trabecular structure with a lamellar suborganization and no cells or cellular remnants were detected. Thus, the bone block was composed of the purified extracellular bone matrix together with a few remnants of the intertrabecular connective tissue. Altogether, the purification level of the analyzed block seems to be favorable as the intact calcified bone matrix together with some tissue-specific collagen were found. Based on these new results, it can be assumed that the level of purification as well as the preservation of the bone matrix in case of the maxgraft® bone ring enables to support the process of integration within bone tissue and bone tissue regeneration. 

## 5. Conclusions

The presented cases demonstrate that the Bone Ring Technique with the allogenic graft is an appropriate procedure for bone augmentation and immediate implant placement in its main indications: three-dimensional ridge reconstruction as well as sinus floor elevation procedure with a maxillary bone height of less than 4 mm. In all presented cases, the use of a FDBA scaffold in a cylindrical shape was a reliable treatment concept. The bone blocks were well regenerated and a second GBR at the time of prosthetic rehabilitation was not necessary. Therefore, a successful and predictable reconstruction of the buccal bone with the cylindrical-shaped bone block can be presumed over a period of 3 years. Altogether, the cases illustrate the benefits of FDBA for the regeneration of alveolar bone and present FDBA as a reliable alternative to harvesting autogenous bone with some advantages: saving valuable surgical time, avoiding donor site morbidity, and finally increasing patient satisfaction. 

Moreover, the material analyses showed that the purification level of the allograft block is favorable as the “intact” calcified bone matrix together with some tissue-specific collagen were found. Based on these new results, it can be assumed that the level of purification as well as the preservation of the bone matrix in case of the maxgraft® bone ring enables to support the process of integration within bone tissue and bone tissue regeneration. 

## Figures and Tables

**Figure 1 materials-12-01291-f001:**
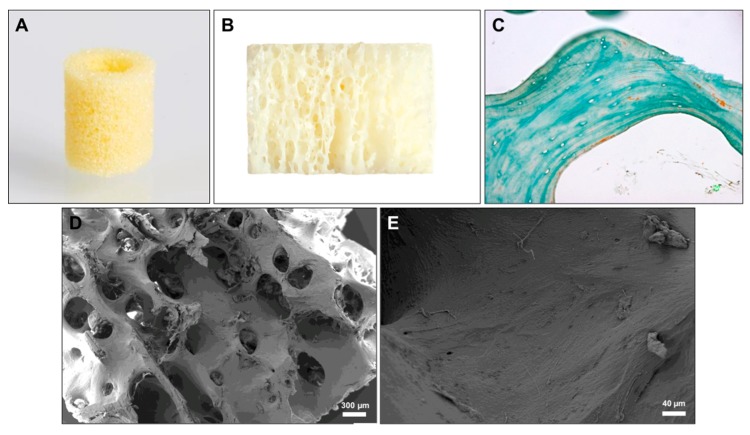
(**A**) Image of the bone ring that shows its porous structure. (**B**) Profile of the bone ring showing the trabecular subarrangement. (**C**) Exemplary histological image of the former bone matrix indicating its lamellar architecture and its complete clearance from cells as well as the matrix-associated collagen fibers (arrow) (Masson Goldner staining, 400x magnification, scale bar = 20 µm). (**D**) Exemplary SEM image that shows the lamellar structure of the bone substitute material and the macro- and micro-porosity (40x magnfication). (**E**) SEM image of the material surface that shows the clearance of the osteocyte lacunae and the fibre-like surface pattern (200x magnification).

**Figure 2 materials-12-01291-f002:**
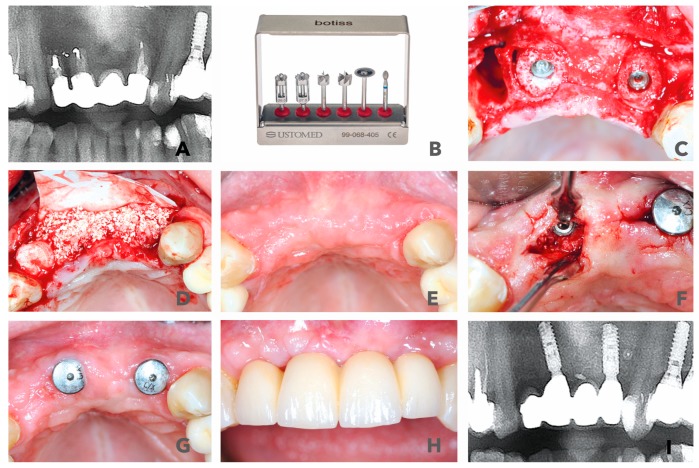
(**A**) Radiological display of the bone defect in frontal maxilla #11-22. (**B**) maxgraft bonering surgical kit. (**C**) Bone augmentation with two cylindrical FDBAs and fixation with dental implants. (**D**) Augmentation site contoured with bovine bone substiute and covered with collagen membrane. (**E**) and (**F**) Six-month follow-up of contouring and re-entry to place gingiva formers. (**G**,**H**) Integration of prosthetic restoration 6 weeks after re-entry and 7.5 months after initial surgery. (**I**) Thirty-six-month radiological follow-up showed stable bone in augmented area.

**Figure 3 materials-12-01291-f003:**
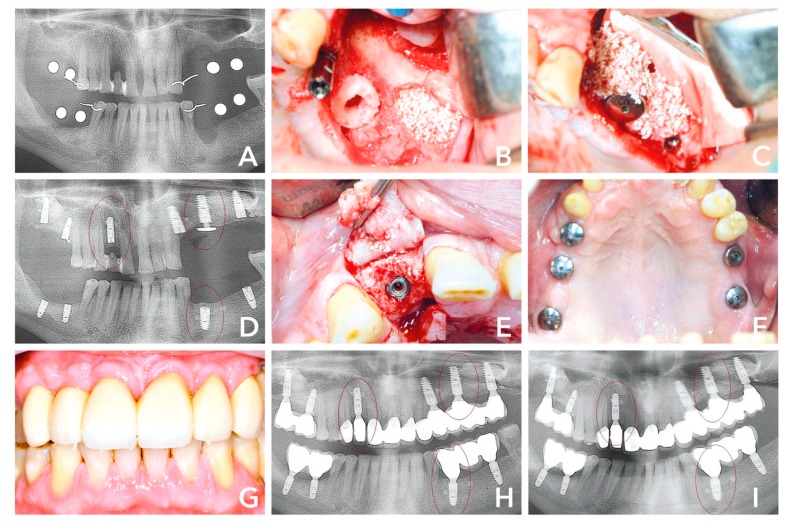
(**A**) Initial situation 6 weeks after extraction of teeth #12, #24-25 and #34. (**B**) Second quadrant: implant placement #28, sinus floor elevation and bone augmenation with a 7-mm bone ring block and immediate implant placement #26. (**C**) Sinus cavity filled with bovine bone substiute and covered with a collagene membrane. (**D**) Radiological control 8 months after surgery with stable bone around the shouders of the implants. (**E**) After opening the flap, the graft seemed to be well integrated. (**F**) Five weeks after re-entry and 9 months after initial surgery healthy soft tissues was observable. (**G**) Prosthtetic restoration in place. (**H**) Radiological control 14 months after surgery, (**I**) compared to 36 months, showed hard tissue maturation and stable dental implants.

**Figure 4 materials-12-01291-f004:**
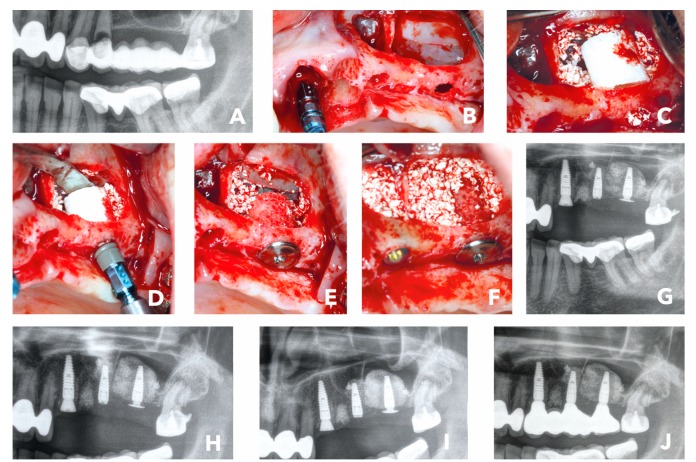
(**A**) Initial situation: Radiological image of bridge retained second quadrant with hopeless teeth #23 and #24. (**B**) External sinus floor elavation and implant placement #23. (**C**) Placement of a 7-mm bone ring block #27 and bovine bone substitute. (**D**) Implant placement. (**E**) Crestal placement of a fication screw to secure the bone graft and implant. (**F**) Implant placement in #25 and sinus cavity filled with bovine bone substitute. (**G**) Post-op x-ray. (**H**) Radiographical control 6 months after surgery. (**I**) Radiographical control 1 week later when the implants were uncovered and prosthodontic impression made. (**J**) Radiographical image of final prosthetic restoration 6.5 months after initial surgery.

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
