# Peer review of "Bone Augmentation and Simultaneous Implant Placement with Allogenic Bone Rings and Analysis of Its Purification Success"

_materials, 2019, doi:10.3390/ma12081291_

Round 1

Reviewer 1 Report

The present article deals with the use of freeze-dried bone allografts (FDBA) by means of a technique of The present article deals with the use of freeze-dried bone allografts (FDBA) by means of a technique of simultaneous bone augmentation and implantation placement ("Bone Ring Technique") The manuscript describes the benefits of the Bonering Technique and the FDBA bone substitute ring on the basis of radiographic and clinical images derived from three different patients.

It is a completely clinic article. The journal Materials, as its name indicates, includes papers which advance the in-depth understanding of the relationship between the structure, the properties or the functions of all kinds of materials.

Although the article has been sent to the biomaterials section, it should have /inlcude a samall characterization  of the material implated, like, physical and chemical properties, microstructure, phase composition, new bone formation, etc. It is therefore a clinical article that discusses the surgical technique, and the clinical results of patients using clinical images. The material is not discussed at any time.

I ask that the article be transferred to a clinical journal for later review.

Author Response

Dear reviewer,

many thanks for the useful comment.

Now we included detailed information about the used biomaterial so that the combination of the material details and the clinical results should be favorablealso for this journal.

We hope this revision and answer and finally the manuscript is suitable for publication.

Many thanks and best wishes,

Mike Barbeck

Reviewer 2 Report

Even if the Authors presented a case report,

The manuscript topic is actual and the paper has merit. It could be attractive, adequate and interesting for the journal readers. However there are some point that authors should address in order to have a final more complete paper. Authors should underline the limitation of the value of the study, and the clinical and surgical implication of the presented cases should be added. At this stage the paper seems to be directed to surgeons and not researchers. Please emphasize the clinical application of the study, and its scientific rationale.

Paper is missed for all the images!!!!!!

References are inadequate. Introduction section is poor. Some more references about the recent (2013-2018) CLINICAL reconstructive option just published recently have to be added. Please add the following ones:

Cicciù, M.; Cervino, G.; Herford, A.S.; Famà, F.; Bramanti, E.; Fiorillo, L.; Lauritano, F.; Sambataro, S.; Troiano, G.; Laino, L. Facial Bone Reconstruction Using both Marine or Non-Marine Bone Substitutes: Evaluation of Current Outcomes in a Systematic Literature Review. Mar. Drugs 201816, 27.

Poli, Pier P et al. “Alveolar ridge augmentation with titanium mesh. A retrospective clinical study” 

open dentistry journal vol. 8 148-58. 29 Sep. 2014, doi:10.2174/1874210601408010148

At the same time discussion is poor.

In the discussion section authors should compare the results of the present study with others one presented and published in the literature. Other important bone substitutes material and clinical studies are the following, please add some samples like:

rhBMP-2 applied as support of distraction osteogenesis: A split-mouth histological study over nonhuman primates mandibles (Article)

Herford AS et al. International Journal of Clinical and Experimental Medicine

Volume 9, Issue 9, 30 September 2016, Pages 17187-17194

Author Response

Dear reviewer,

many thanks for the useful comments,

We have revised the whole manuscript also based on your suggestions and hope it is suitable for publication now.

Best wishes

Mike Barbeck

Reviewer 3 Report

Dear Author 

Looks low readability and lack of scientific data. Could you please add the following paper's information in the introduction and discussion heading 

a) Sheikh, Zeeshan, et al. "Biodegradable materials for bone repair and tissue engineering applications." Materials 8.9 (2015): 5744-5794.

b) https://www.sciencedirect.com/science/article/pii/B9780081009611000050?via%3Dihub 

c) https://www.sciencedirect.com/science/article/pii/B9780081021965000112 

Patient case reporting need better improvement in a proper scientific way. 

Author Response

Dear reviewer,

many thanks for the useful comments. We have now revised the whole manuscript and have included some of your suggestions!

We hope the manuscript is now suitable for publication.

Many thanks and best wishes

Mike Barbeck

Round 2

Reviewer 1 Report

I want to thank the authors who have taken my suggestions into account and included in the article a part dedicated to the material and its characterization.

The authors have made a small introduction where they have given a small view of the different silent materials and then focus on freeze-dried bone allografts (FDBA).

In M & M the authors have explained how the material has been obtained and has been purified.

·         Could the authors include arrows or asterisks in figure 1 C to indicate where the collagen fibers are (line 129 "Slight amounts of collagen fibers were found")?.

·         Line 126 could you please give an average of the micro and macropores size?

·         In line 113 authors indicate to use hematoxylin and eosin (HE) and Masson-Goldner's trichrome staining, but in figure 1C there is only a figure of Masson Goldner staining.

·         Can the authors stick the M & M only to the results shown in the article? Otherwise, a figure with hematoxylin and eosin (HE) staining would be appreciated.

·         Line 138 change mnaterial by material

·         Figure 1 D and 1E have the magnification inside the figure, it is not necessary to repeat the magnification in the figure caption. Please delete it from figure caption.

The authors of the paper have clearly done a lot of work, and the results obtained by them are quite interesting. It reads better now.

Author Response

Dear reviewer,

many thanks for your useful comments that we have included in the manuscript now.

Many thanks and best wishes,

Mike Barbeck

Reviewer 2 Report

Authors made excellent job addressing all the reviewer notes and request

Author Response

Many thanks dear colleague

Reviewer 3 Report

look quite interesting now. Fabulous

Author Response

Many thanks dear colleague